# Task-Interaction-Free Multi-Task Learning with Efficient Hierarchical Feature Representation

## ABSTRACT

Traditional multi-task learning often relies on explicit task interaction mechanisms to enhance multi-task performance. However, these approaches encounter challenges such as negative transfer when jointly learning multiple weakly correlated tasks. Additionally, these methods handle encoded features at a large scale, which escalates computational complexity to ensure dense prediction task performance. In this study, we introduce a Task-Interaction-Free Network (TIF) for multi-task learning, which diverges from explicitly designed task interaction mechanisms. Firstly, we present a Scale Attentive-Feature Fusion Module (SAFF) to enhance each scale in the shared encoder to have rich task-agnostic encoded features. Subsequently, our proposed task and scale-specific decoders efficiently decode the enhanced features shared across tasks without necessitating task-interaction modules. Concretely, we utilize a Self-Feature Distillation Module (SFD) to explore task-specific features at lower scales and the Low-To-High Scale Feature Diffusion Module (LTHD) to diffuse global pixel relationships from low-level to high-level scales. Experiments on publicly available multi-task learning datasets validate that our TIF attains state-of-the-art performance.

## CCS CONCEPTS

• **Computing methodologies** → **Scene understanding**; **Hierarchical representations**.

## KEYWORDS

Multi-Task Learning, Dense Prediction, Vision Transformer, Multi-Scale Features

## 1 INTRODUCTION

Multi-task learning (MTL) involves the joint estimation of multiple dense prediction tasks, which facilitating vision-based scene understanding. MTL addresses pixel-level dense estimation tasks such as semantic segmentation [33], depth estimation [9], boundary detection, and saliency detection [15] within a single unified model. This unified approach effectively achieves high performance across multiple tasks by leveraging task-sharing and task-interaction techniques. Consequently, MTL finds widespread application in various computer vision projects, including Robotics [19] and SLAM [28]. Unlike straightforward MTL methods used for image classification

**Unpublished working draft. Not for distribution.**

[27], MTL for dense prediction tasks [8, 32, 38, 40] presents more complexity due to the intricacies of pixel-level classification and regression.

To address the complexity of learning dense prediction multi-tasks and effectively modeling and training such challenging tasks, researchers have explored MTL through the development of specialized architectures [14, 17, 22, 32, 35, 38, 40, 42] and optimization strategies [3, 10, 11, 18]. Architectural design of MTL primarily comprised of two directions, *(i)* encoder-focused MTL [8, 16, 17, 24] and *(ii)* decoder-focused MTL [32, 35]. Encoder-focused MTL methods [8] typically involve a vast number of parameters due to the absence of a shared encoder, requiring separate encoders for each task. Decoder-focused MTL methods [32] achieve high performance by using a shared encoder and a task interaction mechanism. The shared encoder extracts task-agnostic features from an image, which individual decoders convert into task-specific features. The task interaction module then refines these features using multimodal distillation methods.

However, current decoder-focused methods encounter two primary challenges when processing dense prediction tasks.

Firstly, many task interaction modules in decoder-based methods [1, 32, 35, 38, 43, 45] encounter the negative transfer [44] problem due to the uncertainty regarding the similarity between each pixel of the target task and the source tasks. Several research efforts [1, 26] aim to address this issue. For instance, PADNet [32], PSDNet [45], and MTI-Net [26] have introduced local structures to handle task interaction, ensuring only small spatial relationships between tasks. Leveraging self-attention and Vision Transformers (ViT) [4], methods like InvPT [38], DeMT [36], and TaskPrompter [40] design global relationships between tasks. However, they still struggle to accurately model task interaction and fully resolve negative transfer. We believe one major factor to negative transfer is the potential negative influence of one of the irrelevant source tasks on target tasks through task interaction modules. In Figure 1, we illustrate two tasks, semantic segmentation (SemSeg) and surface normal estimation (Normals). We analyze scenes with task interaction (Model 1) and without task interaction (Model 2). In Model 1, both SemSeg and Normals features are involved to refine Normals task features. Consequently, Model 1 influences the updating weights of SemSeg task, thereby affecting both SemSeg and Normals features if task interaction modules are not well-designed. Conversely, both tasks remain unaffected if all source tasks are SemSeg, as depicted in Model 2 of Figure 1. To prove the observation, we implement the comparison experiment between Model 1 and Model 2. As shown in Table 1, global self attention without other tasks obtains better results compared to global task interaction cross multi-tasks. We compare these models based on different task numbers, which all indicate task interaction encounter the negative transferring.

The second challenge faced by current multi-scale decoder-based MTL methods [26, 38, 40] is the significant computational burden

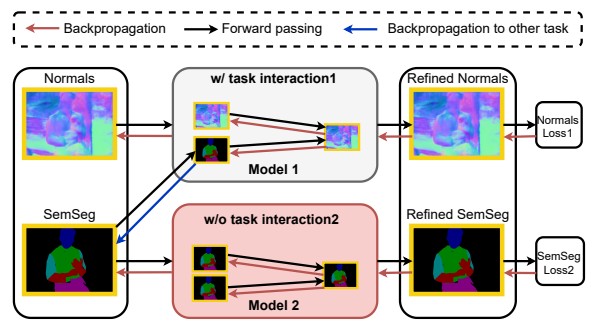

**Figure 1: An example of a factor contributing to negative transfer during task interaction.**

arising from the complexity of pixel-level tasks. For example, designing decoders and task interaction modules at a quarter of input image scale results in high computational resource requirements for pixel-level processing.

Based on these observations, we propose the Task-Interaction-Free Network (TIF) for multi-task learning.

Firstly, instead of designing task interaction modules, we opt to enhance the task-agnostic features from the task-sharing encoder. This decision is motivated by the task sharing encoder typically possesses high number of parameters compared to the decoder. Specifically, the sharing encoder inherently possesses the capability to implicitly achieve task interaction due to its high capacity. Therefore, we propose a novel Scale Attentive-Feature Fusion Module (SAFF) to enhance task-agnostic encoder features at each scale level by utilizing all scale features.

Addressing the second observation for multi-scale decoder-based MTL methods, we introduce two novel modules: *(1)* the Self-Feature Distillation Module (SFD) and the Low-To-High Scale Feature Diffusion Module (LTHD). The SFD module is tasked with extracting self-relative global relationships from task-specific features at the lowest two scales. *(2)* To mitigate complexity at higher scale levels, the LTHD module is proposed to diffuse global relationships from lower to higher levels. This approach ensures that task-specific features encompass both high semantic information and high structural information while maintaining relatively low computational requirements.

In summary, our contributions are as follows:

- We propose the TIF for multi-task learning, comprised of three modules: (1) the Scale Attentive-Feature Fusion Module (SAFF); (2) the Self-Feature Distillation Module (SFD); (3) and the Low-To High Scale Feature Diffusion Module (LTHD). Through integrating these modules, TIF facilitates the processing of multi-tasks without the need for explicit task interaction design.
- We design the SAFF to augment the task-sharing information from the encoder component. This enables the implicit processing of task interaction and facilitates the extraction of rich information from robust encoders.
- SFD and LTHD are proposed to address the computational challenges inherent in current multi-scale based dense prediction multi-tasks.

**Table 1: Performance comparison on model 1 and model 2.**

| Methods | SemSeg (mIoU)↑ | Depth (RMSE)↓ | Normals (mErr)↓ | Bound (odsF)↑ |
|---|---|---|---|---|
| w/ task interaction [1] | 34.67 | 0.6383 | - | - |
| w/o task interaction | **34.78** | **0.6273** | - | - |
| w/ task interaction [1] | - | 0.6346 | 21.19 | - |
| w/o task interaction | - | **0.6292** | **21.06** | - |
| w/ task interaction [1] | **34.90** | 0.6412 | 21.15 | 76.34 |
| w/o task interaction | 34.43 | **0.6247** | **21.11** | **76.57** |

## 2 RELATED WORK

**Multi-task learning for dense prediction**. Multi-task learning (MTL) architectures are primarily categorized into two paradigms: encoder-focused MTL approaches and decoder-focused MTL approaches [1, 32, 35, 38, 39, 43, 45]. Encoder-focused MTL methods have demonstrated efficacy through the deployment of sophisticated encoder-decoder networks [8, 16, 17, 24]. these methods often struggle to improve performance without incorporating task-specific feature interactions at higher feature levels. In contrast, decoder-focused MTL methods have distinguished themselves in multi-task dense prediction scenarios by emphasizing task interaction modules, seeking to excel with a shared encoder complemented by task-specific decoders [25]. PAD [32], a pioneering method in this domain, leverages interactions between auxiliary and target tasks to achieve superior performance on the latter in a localized manner. MTI-Net [26] introduces a novel multi-scale task-specific feature interaction mechanism using a multi-modal distillation module [32] at each scale. ATRC [1] has recently set a benchmark in efficient decoder-focused MTL, harnessing the power of Neural Architecture Search [31] to identify optimal contextual interactions between tasks. InvPT [38] and TaskPrompter [40] achieves remarkable performance by integrating multi-scale techniques [26] with the visual transformer (ViT) architecture [4, 12, 23], showcasing the synergistic potential of these complementary methodologies.

However, the task interaction manner of these methods encounters negative transfer problem. Therefore, we proposed TIF to enhance the task-agnostic features from the encoder and hierarchical task-specific features from decoders, without explicit task interaction.

**Vision Transformer**. The attention mechanism has become increasingly prevalent in computer vision tasks [7], especially in the wake of the transformative impact of Transformers in natural language processing (NLP). This mechanism [13] has outperformed conventional convolutional neural network (CNN)-based methods in numerous dense prediction tasks by leveraging its strength in capturing pixel-level relational information. For dense prediction tasks, a multitude of Vision Transformer (ViT)-based approaches have been developed, each introducing unique self-attention strategies to encapsulate aggregated features across various scales [34], global spatial relationships [30], channel-wise dependencies [6], local contextual information [12], learnable attention weights [41], and focal attention mechanisms [37]. These self-attention methods are primarily designed for individual vision tasks and have not been universally adapted for multi-task learning scenarios. Multi-task

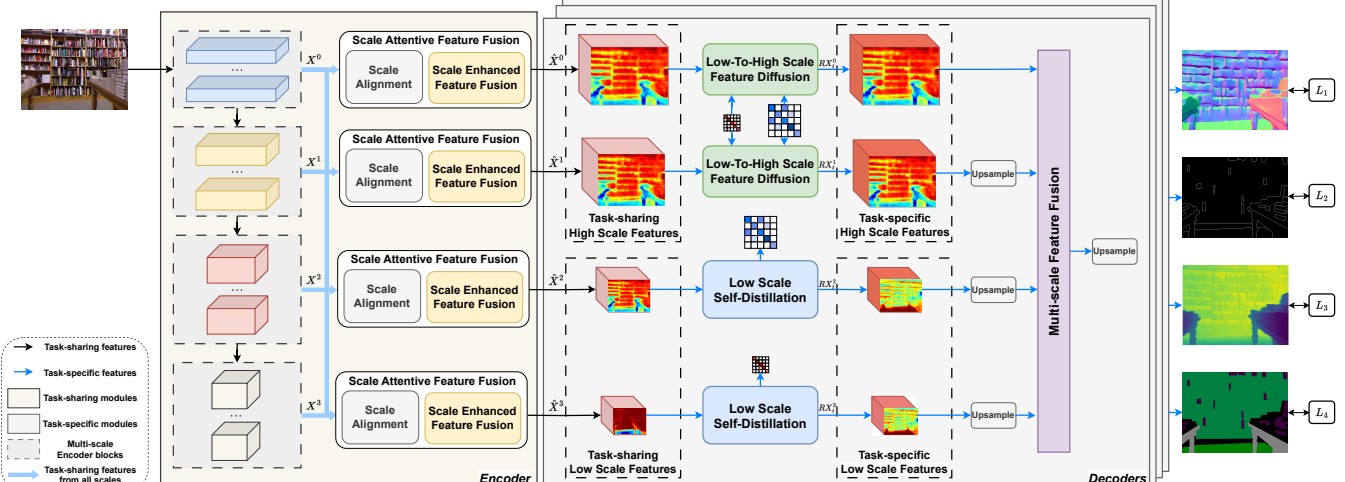

**Figure 2: An overview of the proposed Task-Interaction-Free Network (TIF). TIF consists of a task-sharing encoder and $N$ task-specific decoders for all tasks. In the encoder, Scale Attentive-Feature Fusion Modules (SAFF) generate rich multi-scale task-agnostic features $\hat{X}^0$, $\hat{X}^1$, $\hat{X}^2$, $\hat{X}^3$. In each decoder, Self-Feature Diffusion modules (SFD) model global relationships at the low scales, while Low-To-High Scale Feature Diffusion modules (LTHD) diffuse the global relationships to the high scales. Finally, all low and high scales are fused to generate the final prediction.**

learning methods [36, 38] also utilize the cross-task attention to model global relationships between tasks.

## 3 METHODS

### 3.1 Overview

The proposed framework is composed of a task-sharing encoder in conjunction with $N$ task-specific decoders, as shown in Figure 2. The task-sharing encoder incorporates two principal modules: a feature extractor and the Scale-Attentive Feature Fusion Module (SAFF). The SAFF is further subdivided into two sub-modules: Scale Alignment (SA) and Scale Enhanced Feature Fusion (SEFF). The feature extractor can be any pre-trained vision model [4, 12, 29] with four blocks. Given an image $I \in R^{H \times W \times C}$, the feature extractor generates four scale features:

$$X^0, X^1, X^2, X^3 = FEX(I), X^i \in R^{H/2^{i+2} \times W/2^{i+2} \times C^i} \quad (1)$$

where $X^i$ represents $i$th scale level task sharing feature. $FEX$ denotes the feature extractor function. To refine each scale level $X^i$, we process it through the SAFF modules in conjunction with all four scale features. Initially, the SA modules are employed to align all scale features to the target scale, which is equivalent to the scale of $X^i$. Subsequently, SEFF modules are utilized to augment the target scale feature with the aligned multi-scale features, thereby producing a scale-enhanced feature, denoted as $\hat{X}^i$. As a result, the encoder component generates a rich set of multi-scale task-sharing features, encompassing $\hat{X}^0$, $\hat{X}^1$, $\hat{X}^2$, $\hat{X}^3$.

In contrast to existing decoder-focused MTL methods, our approach is characterized by the design of efficient and effective task-specific decoders for the target tasks. The decoder component of our framework comprises two innovative modules: the Self-Feature

**Table 2: Parameter usage on different backbones.**

| Methods | Backbone | Encoder (M) | Decoders (M) |
|---------|----------|-------------|--------------|
| MQT [35] | Swin-T | 27.52 | 7.83 |
| DeMT[36] | Swin-T | 27.52 | 4.55 |
| MQT [35] | Swin-S | 48.84 | 7.83 |

Distillation Module (SFD) and the Low-To High Scale Feature Diffusion Module (LTHD). Within our framework, we assign $\hat{X}^2$ and $\hat{X}^3$ as the low-scale features and $\hat{X}^1$ and $\hat{X}^0$ as the high-scale features. The SFD modules are introduced to exploit long-range global relationships among these features in a novel and efficient manner. The SFD modules yield two principal outputs: task-specific low-scale features and intra-task affinities. For the high-scale features, the LTHD module is introduced to facilitate the diffusion of intra-task affinities from the low-scale to the high-scale levels. Ultimately, the task-specific high-scale features and low-scale features are concatenated to produce the final task-specific features, which are then utilized for the respective task predictions.

### 3.2 Scale Attentive Feature Fusion

In the realm of multi-task learning, it is a common observation that the parameters of the encoder are significantly larger than those of the decoders, as illustrated in Table 2. Moreover, encoders are typically pre-trained on extensive image datasets, endowing them with a wealth of image representation information compared to the more lightweight decoders. In light of this, we introduce the Scale Attentive Feature Fusion Module (SAFF) to harness the rich information embedded within the task-sharing feature extractor (Figure 3). The SAFF module comprises two key sub-modules: Scale

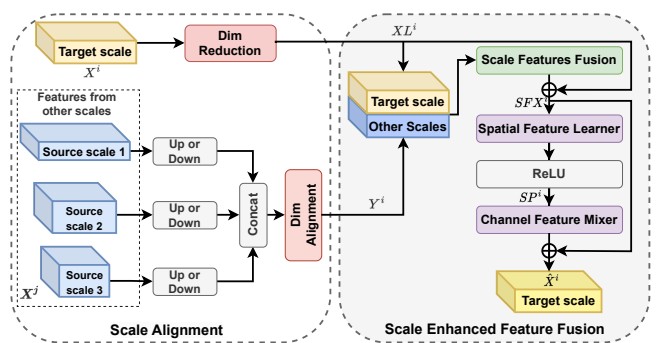

Figure 3: Illustration of Scale Attentive-Feature Fusion module.

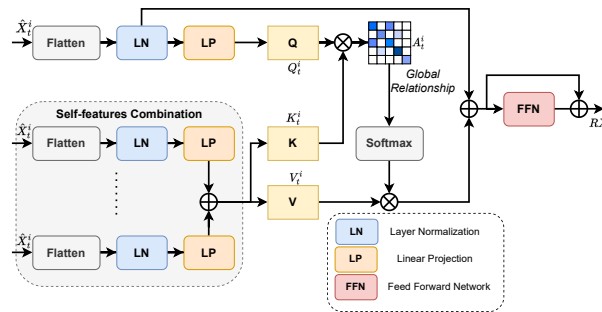

Figure 4: Illustration of Self-Feature Distillation module.

Alignment (SA) and Scale Enhanced Feature Fusion (SEFF). As depicted in our framework (Figure 2), all high and low-level features $(X^0, X^1, X^2, X^3)$ are refined by the SAFF modules.

**Scale Alignment.** Given a task-sharing feature $X^i$ at the target scale level $i$, SA modules align all other source scale features to the same scale with the scale $i$:

$$Y^i = W_i \times f_{cat}\{\mathcal{P}_{H^i \times W^i}(X^j)\}, j \neq i, \quad (2)$$

where $Y^i$ denotes the $i$th scale feature that contains all scale features except scale $i$. $X^j$ denotes a feature set contains three source scales. $\mathcal{P}_{H^i \times W^i}$ denotes that changes input scales to the same scale with $X^i$. $f_{cat}$ denotes the concatenation on the channel size. $W_i$ denotes the reshape dimension operation with learnable weights. To avoid high computational problems, we reduce the dimension of $X^i$ to the low dimension $XL^i$ with a $1 \times 1$ convolution and batch normalization. $W_i$ reduces the dimension of the other scale information $Y^i$ to the same dimension with $XL^i$, otherwise high dimension of $Y^i$ lead to model losing the target scale information.

**Scale Enhanced Feature Fusion.** SEFF module incorporates the information from other scales $Y^i$ to the target scale $X^i$. In this way, target scale features contain rich semantic and structural information from other scales. The architectural design of the SEFF module is inspired by the structure of the Transformer block. The inclusion of convolution to capture more nuanced structural information, as convolution has been observed to excel in this regard compared to attention mechanisms. In our approach, unlike conventional Transformer-based methods, we fusion the benefits of both the Transformer block and convolution. Firstly, given the feature $XL^i$ from scale $i$ and $Y^i$ from other scales, SEFF learns a a unified feature representation for scale $i$ with a Scale features fusion module. Subsequently, to preserve the integrity of the information specific to the target scale $i$, $XL^i$ are added to the fused feature:

$$SFX^i = SFF(XL^i, Y^i) + XL^i, \quad (3)$$

where $SFF()$ is the Scale features fusion function. In this paper, we utilize two convolution layers and ReLU activation function to achieve $SFF()$. With this function, we obtain well-learned scale fused feature $SFX^i$ with rich information from target scale $i$ and other scales. Subsequently, we enhance the feature learning process by introducing a dedicated spatial feature learner and a channel

feature mixer, which collectively refine both the spatial and channel dimensions for scale $i$. For the spatial feature learner, we have crafted a group-wise $3\times3$ convolution, which is specifically aimed at capturing local spatial structures. In parallel, for the channel feature mixer, we utilize a $1 \times 1$ convolution to adeptly blend and refine the channel-wise information derived from the spatial features. This dual approach ensures a comprehensive learning process that attend to both the spatial and channel aspects of the feature representation, thereby enhancing the overall discriminative power of the scale-specific features.

$$SP^i = ReLU(f_{group}(SFX^i)), \quad (4)$$

$$\hat{X}^i = f_{1\times1}(SP^i), \quad (5)$$

where $SP^i$ denotes $i$th scale enhanced spatial features. $f_{group}$ denotes group-wise convolution with a group equal to the input channel number. $f_{1\times1}()$ means $1 \times 1$ convolution function for mixing channel. $\hat{X}^i$ represents the $i$th scale feature with enhanced by the other scale levels. As a result, $\hat{X}^i$ is enriched with information not only from scale $i$ but also from the other scales, which collectively offer a broad spectrum of semantic and spatial insights from the shared feature extractor. To amplify the characteristics of each target scale feature, we design a SEFF module for each individual scale. Through the application of these specialized modules, we procured four sets of enhanced scale features: $\hat{X}^0, \hat{X}^1, \hat{X}^2, \hat{X}^3$. These comprehensive feature sets lay the foundation for the robust multi-task learning capabilities of our framework.

## 3.3 Self-Feature Distillation for Low Scales

Traditional dense prediction methods [1, 36] process high scale features on the decoder part. The scale of the highest level feature is generally $H/4 \times W/4$, while the lowest level feature is $H/32 \times W/32$. The complexity of the highest scale is precisely **64 times** that of the lowest when self-attention is applied to them. Additionally, the inter-pixel relationships at the highest scale level often yield global relationships that are predominantly zero. This suggests that high scales may predominantly encode structural information rather than global relationships. Hence, we introduce a Self-Feature Distillation Module (SFD) to extract global relationships from the lowest two scale features, as depicted in Figure 4.

Unlike current decoder-focused MTL methods, the input to each SFD is restricted to the low-scale target task-specific features, excluding any other source task features. Consequently, SFD modules

**Table 3: Comparison results on NYUD-v2 dataset. '↑' means higher is better and '↓' means lower is better.**

| Type | Method | Backbone | Param(M) | GFLOPS(G) | SemSeg(mIoU)↑ | Depth(RMSE)↓ | Normals(mErr)↓ | Bound(odsF)↑ |
|---|---|---|---|---|---|---|---|---|
| Single-Scale | Cross-Stitch [16] | HRNet18 | **4.52** | 17.59 | 36.34 | 0.629 | 20.88 | 76.38 |
| | PAD-Net [32] | HRNet18 | 5.02 | 25.18 | 36.7 | 0.6264 | 20.85 | 76.5 |
| | NDDR-CNN [8] | HRNet18 | 4.59 | 18.68 | 36.72 | 0.6288 | 20.89 | 76.32 |
| | PAP [43] | HRNet18 | 4.54 | 53.04 | 36.72 | 0.6178 | 20.82 | 76.42 |
| | PSD [45] | HRNet18 | 4.71 | 21.1 | 36.69 | 0.6246 | 20.87 | 76.38 |
| | Global-context [1] | HRNet18 | 4.73 | 21.43 | 38.3 | 0.6007 | 20.6 | 76.26 |
| | ATRC [1] | HRNet18 | 5.06 | 25.76 | 38.9 | 0.601 | 20.48 | 76.34 |
| | DeMT [36] | HRNet18 | 4.76 | 22.07 | 39.1 | **0.5922** | **20.21** | 76.4 |
| Multi-scale | MTI-Net [26] | HRNet18 | 12.56 | 19.14 | 36.61 | 0.627 | 20.85 | 76.38 |
| | SATMTN [21] | HRNet18 | 5.41 | 24.69 | 38.92 | 0.5952 | 20.29 | 76.77 |
| | **TIF (ours)** | HRNet18 | 5.27 | **16.6** | **40.28** | 0.594 | 20.34 | **76.92** |
| Single-Scale | DeMT [36] | Swin-T | 32.07 | 100.7 | 46.36 | 0.5871 | 20.65 | 76.9 |
| Multi-scale | InvPT [38] | Swin-T | - | - | 44.27 | **0.5589** | 20.46 | 76.10 |
| | MQTransformer [35] | Swin-T | 35.35 | 106.02 | 43.61 | 0.5979 | **20.05** | 76.90 |
| | **TIF (ours)** | Swin-T | 32.91 | 76.33(24.37↓) | 47.42 | 0.5677 | 20.11 | **77.75** |

do not involve task interaction processes. We abuse $\hat{X}_t^i$ to be the task $t$ feature at scale $i$. To extract the intra-task global relationships from the low-scale features, we incorporate the self-attention mechanism within the SFD modules. Instead of gathering information from other tasks, we focus on extracting diverse information from the target task itself. Consequently, the key and value are derived by self-feature combination, with a set of target task-specific features serving as the input. Given a task-specific feature $\hat{X}_t^i, i \in \{2, 3\}$:

$$K_t^i = V_t^i = Flat(\hat{X}_t^i)W_0^i + Flat(\hat{X}_t^i)W_2^i + \cdot + Flat(\hat{X}_t^i)W_N^i, \quad (6)$$

$$Q_t^i = Flat(\hat{X}_0^i)W_{tq}^i, \quad (7)$$

$$A_t^i = Q_t^i \times K_t^i, \quad (8)$$

where, $Q, K, V \in R^{H^i W^i \times C}$ denote the query, key, and value of the SFD module. K is equal to V in our setting for the model efficiency. Flat() function denotes the flatten and linear normalization process to the input feature map. $N$ is the task number and $W$ means the learnable weights. Therefore, K and V are the combination of various learned features from the $t$th target task feature itself. $A_t^i \in R^{H^i W^i \times H^i W^i}$ denotes the global relationships of task $t$ from the scale $i$. In this way, $A_t^i$ from the low scales $2, 3$ contains rich long-range global relationships, which is designed to diffuse these

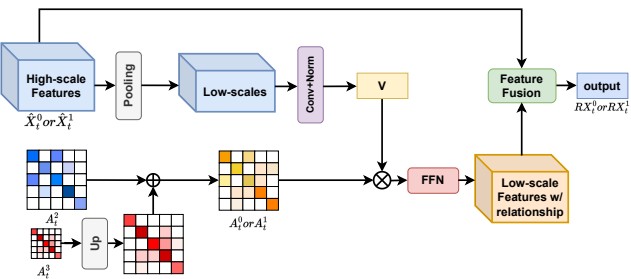

**Figure 5: Illustration of Low-To-High Scale Feature Diffusion module.**

relationships to the higher scales later, as illustrated by **subsection 3.4**. Then, we obtain the refined low scale task feature $RX_t^i$ by:

$$RX_t^i = Softmax(A_t^i) \times V_t^i + \hat{X}_t^i, \quad (9)$$

$$RX_t^i = FFN(RX_t^i) + RX_t^i \quad (10)$$

where FFN() denotes the feed-forward network. With residual process with $\hat{X}_t^i$, $RX_t^i$ also contains the information from the encoder part. SFD modules save the computational resources with self-attention way on the low task-specific feature scales. Moreover, SFD modules without cross-task information do not encounter the problem mentioned from Figure 1 in **section 1** , without using task interaction. Finally, we process SFD modules to the low-scales features to obtain refined task-specific features $RX_t^2$ and $RX_t^3$.

### 3.4 Low-To-High Scale Feature Diffusion

Typically, high-scale features encapsulate rich structural information, whereas low scales are characterized by high-level semantic relationship information. Current multi-task learning methods [1, 35, 36] typically process high-scale features that integrate information from both high and low levels. This approach embodies a rich mixture of semantic and structural information. However, the model complexity for high-scale features is significantly greater than that of low scales. Therefore, we introduce the Low-To High Scale Feature Diffusion (LTHD) module to propagate the low-scale global relationships to the high scales, as depicted in Figure 5.

Given the global relationships $A_t^2$ and $A_t^3$, the objective of LTHD modules is to propagate these relationships to the high-scale features $\hat{X}_t^0$ and $\hat{X}_t^1$ for task $t$. The scale size of the high-level features is substantially larger than that of the lower scales. Therefore, processing the diffusion from $A_t^2$ and $A_t^3$ to the high scales in the high scale level increases the model complexity. Furthermore, most of the global relationships between pixels on the high scales are close to zero, necessitating a more refined approach to information diffusion. To address these challenges, in LTHD, we first process the pooling operation to the high-scale features $\hat{X}_t^0$ and $\hat{X}_t^1$ to produce a lower-scale version of them. Since the scale levels of the global relationships $A_t^2$ and $A_t^3$ are are distinct, we upsample $A_t^3$ to match the scale with $A_t^2$. Subsequently, we employ a cross-scale attention

**Table 4: Performance comparison with large parameter multi-scale based SOTA methods on NYUD-V2 dataset.**

| Methods | SemSeg (mIoU)↑ | Depth (RMSE)↓ | Normals(mErr)↓ | Bound (odsF)↑ | backbones | Parameters (M) | Flops (G) |
|---|---|---|---|---|---|---|---|
| InvPT[38] | 53.56 | 0.5183 | 19.04 | 78.10 | Vit-L | 402.1 | 555.57 |
| MQTransformer[35] | 54.84 | 0.5325 | 19.67 | 78.20 | Swin-L | 204.3 | 365.25 |
| TaskPrompter[40] | 55.30 | 0.5152 | **18.47** | 78.20 | Vit-L | 392* | 470* |
| TaskExpert[39] | 55.35 | 0.5157 | 18.54 | 78.40 | Vit-L | 392+ | 470+ |
| **TIF (ours)** | **56.80** | **0.5023** | 19.21 | **79.52** | Swin-L | **204.19** | 274.27 (**90.98↓**) |

mechanism to diffuse the integrated global relationships to the high scales. Specifically, for the scale 0 features of task $t$ $\hat{X}_t^0$:

$$A_t^0 = f_{up}(A_t^3) + A_t^2, \tag{11}$$

$$RX_t^0 = Pool(\hat{X}_t^0)W_t^0 \times Softmax(A_t^0), \tag{12}$$

$$RX_t^0 = FeatureFuse(FFN(RX_t^0), \hat{X}_t^0), \tag{13}$$

where, $A_t^0$ represents the approximate attention map for scale 0 of task $t$. Instead of employing self-attention on high scales to generate $A_t^0$, we design it with low-scale global relationships. $RX_t^0$ denotes the refined features from scale 0. $FeatureFuse()$ signifies the fusion process using Convolutional layers followed by Batch Normalization. By processing the LTHD modules on the highest two scales, we obtain refined task-specific features $RX_t^0$ and $RX_t^1$. Finally, the ultimate task-specific features $FX_t$ are obtained by:

$$FX_t = FeatureFuse(RX_t^0, RX_t^1, RX_t^2, RX_t^3). \tag{14}$$

The loss function of the proposed framework is the combination of all task loss functions with different fixed weights. We follow the current methods [1] to design the overall loss function.

## 4 EXPERIMENTS

### 4.1 Dataset

The **PASCAL-Context** dataset [2] is an extension of the PASCAL VOC dataset [5]. It comprises 10,103 images, with 4,998 designated for training and the remaining 5,105 for testing. This dataset is annotated for five distinct tasks: boundary detection, surface normal estimation, semantic segmentation, human part segmentation, and saliency estimation. The **NYUD-v2** dataset [20] focuses on indoor scenes and includes 1,449 labeled RGB images. Of these, 795 are randomly allocated for training, while 654 are reserved for testing. Each image in NYUD-v2 is annotated for four dense prediction tasks: semantic segmentation, depth estimation, boundary detection, and surface normal estimation.

### 4.2 Implementation Details and Metrics

**Implementation Details**. To ensure a fair comparison, we use identical experimental settings as other decoder-focused MTL methods [1]. Our backbone model employs HRNet18 [29], Swin Transformer [12] and Vit-B. For HRNet18 and Swin Transformer, we set the learning rate to 0.01 and 0.001 for the NYUD-v2 dataset, and 0.0001 for the PASCAL-Context dataset. Across all experiments, we maintain a batch size of 8 and train for 50k iterations with a weight decay rate of 0.0005. All training processes are conducted on NIVDIA RTX3090 GPUs. More details will be provided by the source code.

**Metrics**. We follow the existing MTL methods [1, 35, 39] to evaluate the performance of semantic segmentation (SemSeg) and human parts segmentation (PartSeg) using the mean Intersection over Union (mIoU) metric. For depth estimation (Depth), the root mean square error (RMSE) is utilized, and saliency detection (Sal) is measured by the maximal F-measure (maxF). Surface normal estimation (Normals) is assessed using the mean angular error (mErr), and boundary detection (Bound) employs the optimal-dataset-scale F-measure (odsF). We evaluate the multi-task performance with the per-task performance drop ($\Delta_m = \frac{1}{N} \sum_{i=1}^N \frac{(P_m^i - P_{base}^i)}{P_{base}^i}(-1)^{w_i}$), where $P_m^i$ denotes the performance of MTL method on the $i$-th task and $P_{base}^i$ denotes the baseline performance of this task. $(-1)^{w_i}$ denotes the sign symbol, $(-1)^{w_i} = 1$ if higher value performance is better, and $(-1)^{w_i} = -1$ if lower value performance is better.

### 4.3 Comparison with the state-of-the-arts

We implement the comparison experiments with existing MTL methods [1, 32, 35, 38, 39, 43, 45] on both multi-task datasets NYUD-v2 and PASCAL-Context.

**NYUD-v2**. As shown in Table 3, the proposed TIF surpasses the current methods both on CNN and Transformer backbones without using any explicit task interaction methods. Meanwhile, the model complexity of the proposed method also achieves better performance with the current efficient single-scale decoder-based MTL methods such as DeMT [36]. It indicates that task-specific decoders with the proposed SFD and LTHD modules can achieve high accuracy on multi-tasks as well as high efficiency. Furthermore, as shown in Table 4, we compare our method with the large-scale MTL methods (multi-scale decoder-based) such as TaskPrompter [40], TaskExpert [39], InvPT [38], MQT [35]. The results show that the proposed TIF achieves SOTA performance with small parameters and FLOPs. **PASCAL-Context**. We further evaluate our works with the existing methods on the PASCAL-Context dataset. As shown in 5, our method surpasses the current MTL methods both on the efficient CNN and effective Transformer backbones. Particularly, the performance of our method surpasses DeMT **2.95%, 2.86%** on the PartSeg task with Swin-B and Swin-T backbones. It also surpasses DeMT on Bound task **2.42%, 2.17%** on Swin-B and Swin-T.

The comparison results on both two public datasets and Table 1 denotes that the task interaction technique used by current methods still has negative transfer problems. The proposed TIF with a simple encoder-decoder design achieves state-of-the-art performance without using task interaction modules. Instead, we replace it by strengthening the task interaction by enhancing the task-sharing encoder, which is also demonstrated by Table 1. By designing efficient decoders without explicit task interaction, the model complexity

**Table 5: Comparison Results on PASCAL-Context dataset.**

| Methods | Backbone | SemSeg(mIoU)↑ | PartSeg(mIoU)↑ | Sal(maxF)↑ | Normals(mErr)↓ | Bound(odsF)↑ |
|---|---|---|---|---|---|---|
| ATRC [1] | HRNet18 | 57.89 | 57.33 | 83.77 | 13.99 | 69.74 |
| MQTransformer [35] | HRNet18 | 58.91 | 57.43 | 83.78 | 14.17 | 69.8 |
| DeMT [36] | HRNet18 | **59.23** | **57.93** | 83.93 | 14.02 | 69.80 |
| **TIF (ours)** | HRNet18 | 58.24 | 57.92 | **84.55** | **13.91** | **70.89** |
| MQTransformer [35] | Swin-T | 68.24 | 57.05 | 83.40 | 14.56 | 71.10 |
| DeMT [36] | Swin-T | **69.71** | 57.18 | 82.63 | 14.56 | 71.20 |
| **TIF (ours)** | Swin-T | 69.22 | **60.04** | **84.05** | **13.92** | **73.37** |
| DeMT [36] | Swin-B | **75.33** | 63.11 | 83.42 | 14.54 | 73.20 |
| **TIF (ours)** | Swin-B | 74.89 | **66.06** | **84.30** | **13.80** | **75.62** |

**Table 6: Effectiveness of proposed modules.**

| SAFF | SFD | LTHD | SemSeg (mIoU)↑ | Depth (RMSE)↓ | Normals (mErr)↓ | Bound (odsF)↑ |
|---|---|---|---|---|---|---|
| ✗ | ✗ | ✗ | 45.42 | 0.5864 | 20.48 | 77.41 |
| ✔ | ✗ | ✗ | 46.56 | 0.5901 | 20.62 | 77.76 |
| ✔ | ✔ | ✗ | 46.71 | 0.5766 | 20.29 | 77.78 |
| ✔ | ✔ | ✔ | 47.42 | 0.5677 | 20.11 | 77.75 |

**Table 7: Effectiveness of proposed SFD modules.**

| SFD | SemSeg (mIoU)↑ | Depth (RMSE)↓ | Normals (mErr)↓ | Bound (odsF)↑ | $\Delta_m$ (%) ↑ |
|---|---|---|---|---|---|
| Baseline | 42.98 | 0.6012 | 21.39 | 65.15 | 0.0 |
| W/1 SFD | 46.93 | 0.5741 | 20.20 | 77.81 | 5.70 |
| W/2 SFD | 47.42 | 0.5677 | 20.11 | 77.75 | 6.34 |

**Table 8: Performance comparison between cross-task interaction and self-distillation.**

| Distillation | SemSeg (mIoU)↑ | Depth (RMSE)↓ | Normals (mErr)↓ | Bound (odsF)↑ | $\Delta_m$ (%) ↑ |
|---|---|---|---|---|---|
| Baseline | 42.98 | 0.6012 | 21.39 | 65.15 | 0.0 |
| Cross-distill | 45.79 | 0.5738 | 20.57 | 77.20 | 4.41 |
| Self-distill | 46.71 | 0.5766 | 20.29 | 77.78 | 5.35 |

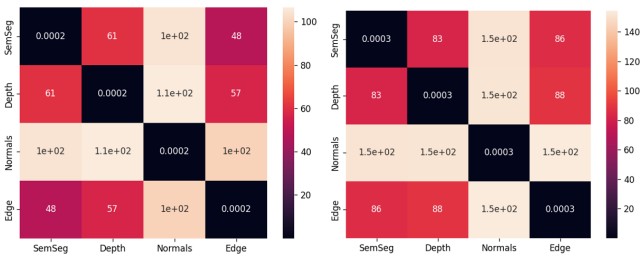

(a) Between-task distance on Scale3    (b) Between-task distance on Scale2

**Figure 6: Between-task distance on Scale3 and Scale2 for the SFD modules.**

surpass remarkably compared to multi-scale decoder-based methods [35, 38–40] and the efficient SOTA methods such as DeMT[36].

## 4.4 Ablation Study

*4.4.1 Effectiveness of the proposed modules.* As reported in Table 6, we conduct an ablation study on our proposed methods with the Swin-T backbone on the NYUD-V2 dataset. The optimal performance is achieved when all modules SAFF, SFD, and LTHD are included. The results indicate that SemSeg and Depth performance notably degrades when LTHD is excluded, underscoring the significance of both the global dependency propagated to the high scales and the local details at those scales for these tasks. SFD extracts the inner-task global relationships, suggesting that tasks like Depth are sensitive to the global dependency at lower scales. The performance of SemSeg notably drops without SAFF, highlighting the importance of the rich information from the task-sharing encoder.

*4.4.2 Effectiveness of the SFD modules.* Table 7 demonstrates that applying SFD modules to the two lowest scales outperforms using it on a single scale. This indicates that the global relationships at both scales are advantageous for the performance of most tasks.

Furthermore, the model complexity is notably reduced when applying SFD to the two lowest scales compared to using it on the higher scales. Consequently, we exclusively utilize SFD on the two lowest scales.

*4.4.3 Analysis on task interaction.* As shown in Table 8, we implement the SFD in two ways: cross-task distillation and self-distillation. Cross-task distillation is widely used by current MTL methods to achieve task interaction globally, while self-distillation means the proposed SFD modules. The results show that self-distillation has better performances compared to the cross-distillation. The results also are consistent with Table 1. Therefore, such task interaction manner still causes negative transfer problems. It also indicates that cross-task messages should be precisely transferred to the target tasks, otherwise, it will encounter negative transfer problems. Besides, the parameter and FLOPs of cross-task distillation are 31.56 M and 62.89 G, while SFD modules take 31.78 M and 62.49 G. It

**Table 9: Effectiveness of proposed SAFF modules.**

| SAFF | SemSeg (mIoU)↑ | Depth (RMSE)↓ | Normals (mErr)↓ | Bound (odsF)↑ | $\Delta_m$ (%) ↑ |
|---|---|---|---|---|---|
| Baseline | 42.98 | 0.6012 | 21.39 | 65.15 | 0.0 |
| W/O SAFF | 46.12 | 0.5678 | 20.03 | 77.34 | 5.53 |
| W/2 SAFF | 46.29 | 0.5675 | 20.03 | 77.34 | 5.64 |
| W/4 SAFF | 47.42 | 0.5677 | 20.11 | 77.75 | 6.34 |

**Table 10: Results on multi-scale decoder-based MTL methods.**

| Methods | backbone | SemSeg (mIoU)↑ | Depth (RMSE)↓ | Normals (mErr)↓ | Bound (odsF)↑ | param |
|---|---|---|---|---|---|---|
| MTI-Net | HRNet18 | 36.61 | 0.6270 | 20.85 | 76.38 | 12.56 |
| **TIF(ours)** | HRNet18 | 40.28 | 0.5940 | 20.34 | 76.92 | **5.27** |
| TaskPromter | Vit-B | 50.40 | 0.5402 | 18.91 | 77.60 | 350* |
| **TIF (ours)** | Vit-B | 52.57 | 0.5360 | 18.78 | 77.16 | **124** |

shows that the proposed SFD modules have the comparable model complexity with the cross-task distillation.

*4.4.4 The importance of SAFF modules.* As shown in Table 9, we implement different numbers of SAFF to different scales. Specifically, W/o SAFF means without using SAFF to the multi-scales. W/2 SAFF means only using SAFF on the lowest two scales. W/4 SAFF means the full model. It shows that W/4 SAFF improves the performance of W/2 SAFF than W/2 SAFF to W/O SAFF. It indicates that multi-scale fusion to the high scales is much better than to the low scales for dense prediction tasks. Therefore, high scales still are important features for dense predictions.

*4.4.5 Analysis on multi-scale MTL methods.* As shown in Table 10, we compare multi-scale MTL methods, assessing model parameters and performance with HRNet18 and Vit-B backbones. TaskPromoter [40] and MTI-Net [26] are SOTA MTL methods based on multi-scale decoders. Our TIF method also follows this approach. TIF outperforms TaskPromoter in performance with a smaller model size, demonstrating remarkable MTL performance without task interaction modules or complex high-scale global and local dependencies.

## 4.5 Qualitative study

In this section, we conduct three qualitative studies on the proposed TIF. Firstly, we conduct task similarity experiments, as illustrated in Figure 6, revealing that task similarity varies across the SFD modules at scales 2 and 3. The lower scale exhibits higher task similarity than the higher scale, validating the introduction of SFD modules at the lowest two scales. Moreover, different tasks exhibit varying task similarities at both scales 2 and 3, with some tasks exhibiting high dissimilarity. This dissimilarity can lead to negative transfer when implementing task interaction. Additionally, we observe that different input images display distinct task similarities, making it challenging to model an explicit and accurate task interaction method for dense prediction multi-tasks. Secondly, we

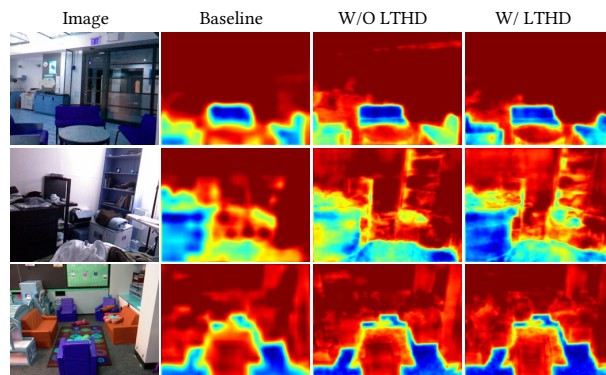

**Figure 7: Feature visualization for LTHD modules.**

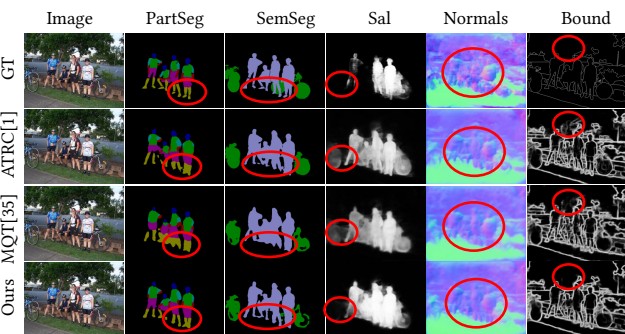

**Figure 8: Visual comparison of the proposed ATMPNet with ATRC [1] and MQT [35] in complex scenes.**

conduct qualitative experiments to underscore the significance of LTHD modules. As depicted in Figure 7, models with LTHD modules achieve more precise predictions for multiple and large objects compared to those without LTHD, indicating that LTHD modules enhance performance by diffusing global relationships in SemSeg tasks. This aligns with the quantitative results presented in Table 6. Thirdly, we compare our method with the current CNN-based approach ATRC [1] and the transformer-based method MQT [35]. As shown in Figure 8, our method demonstrates superior performance on most tasks without relying on task interaction.

## 5 CONCLUSION

In this paper, we introduce a Task-Interaction-Free Network (TIF) for multi-task learning. Unlike traditional methods rely on task interaction modules, TIF leverages a novel task-agnostic SAFF to enhance task interaction by task sharing information the encoder part. This approach avoids the negative transfer commonly encountered in task interaction strategies. Additionally, to mitigate the computational complexity of multi-scale decoder-based MTL methods, we propose two complementary modules: the SFD and LTHD modules. The SFD modules extract rich inner-task global dependencies for low-scales, while the LTHD modules propagate these dependencies from the lower scales to the higher scales.

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
