# OpenReview forum: "Task-Interaction-Free Multi-Task Learning with Efficient Hierarchical Feature Representation"
_acmmm.org/ACMMM/2024/Conference — MM2024 Poster_

### Official Review · Reviewer_k6Rn · 2024-05-24

**Rating:** 4
**Confidence:** 2

**Summary:**

The authors introduce a Task-Interaction-Free Network including three modules to avoid the negative transfer commonly encountered in task interaction strategies.

**Strengths:**

1. Clear motivation.

2. Detailed illustration of the proposed method.

3. Comprehensive experiments, convincing results, and clear presentation.

**Limitations:**

1. Need to discuss and compare with more recent SOTA methods

2. The comparison of number of parameters and FLOPs are not presented in Table 5.

3. Missed “Table” in Line 682.

4. The limitations of the work are not discussed.

5. More analyses on the reasons for the method's effectiveness may be better.

**Suitability:**

2

---

### Official Review · Reviewer_nnEZ · 2024-05-25

**Rating:** 5
**Confidence:** 3

**Summary:**

This paper proposed a task-interaction-free multi-task learning framework with efficient hierarchical feature representation, it mainly introduces 3 components: SAFF module providing rich task-agnostic features, SFD module capturing global semantic relationships and LTHD diffusing global pixel relationships.

**Strengths:**

The strengths are:
(1) They conduct adequate evaluation to show the superiority of the proposed method.
(2) The novelty is good, especially “the negative transfer when jointly learning multiple weakly correlated tasks” described in Figure 1.
(3) The clarity is good; the figures of each component and detailed explanation are easy to understand.

**Limitations:**

The limitations are:
(1) While the existence of negative transfer in multi-task learning is indeed interesting, it is essential to determine when the impact of negative transfer between multiple tasks is so severe that it renders the design of a task-interaction module unnecessary. This question needs to be addressed because it affects the generalizability of the proposed method.
(2) In Table 2, besides the SwinTransformer backbone, are there other backbones used in the multi-task learning model? If so, please provide more multi-task learning model to illustrate that the encoder is heavier than decoder.
(3) In the spatial feature learner of Figure 3, have you tried using a larger kernel size besides 3 by 3?
(4) In Section 3.4 “most of the global relationships between pixels on the high scales are close to zero, necessitating a more refined approach to information diffusion.” could you provide some proof to support this conclusion?
(5) In Formula 7, X ̂_0^i should be X ̂_t^i .
(6) In the Task-specific modules, there are a total of four scales features (X ̂^0, X ̂^1, X ̂^2, X ̂^3). The authors classify the first two scales as high-scale features with more structural information and the last two scales as low-scale features with more semantic information. My question is whether it would be sufficient to use only the first high-scale feature and only the last low-scale feature in SFD and LTHD, and could you describe the complementary between X ̂^0and X ̂^1 in LTHD, and X ̂^2and X ̂^3 in SFD.

**Suitability:**

2

---

### Official Review · Reviewer_2zx2 · 2024-05-26

**Rating:** 3
**Confidence:** 2

**Summary:**

This paper proposes a Task-Interaction-Free Network (TIF) that uses a designed task-agnostic SAFF to model the task-level relation. The authors that the proposed method can alleviate the negative transfer. Besides, the authors propose the SFD and LTHD modules to reduce the computational complexity. Experiments are conducted on PASCAL-Context and NYUD-v2, and show the performance gain when using their method.

**Strengths:**

This paper conducts extensive experiments to show the effectiveness of the proposed method.

This paper is well-organized and easy to follow

**Limitations:**

- The proposed TIF consisting of a task-sharing encoder and 𝑁 task-specific decoders, which is not novel and a common network pipeline for addressing the multi-task problem. The main technical difference is that the authors design a SAFF to encode the multi-scale features. I think that the SAFF also lacks a certain degree of insight. It is suggested that the authors improve it during the rebuttal period.

- This paper introducs many enchanced modules such as Self-Feature Distillation module, Scale Attentive-Feature Fusion module, Low-To-High Scale Feature Diffusion to boost the multi-task performance, which seems to be an incremental improvement and lacks deep insight for multi-task learning community.

**Suitability:**

3

---

### Meta-Review · Area_Chair_6K7W · 2024-07-04

**Recommendation:** Accept (Poster)
**Confidence:** 3

**Metareview:**

- Summary of Strengths and Contributions

The paper introduces a multi-task learning framework aimed at mitigating negative transfer between tasks. The framework includes the Scale Attentive-Feature Fusion (SAFF) module, the Self-Feature Distillation (SFD) module, and the Low-To-High Scale Feature Diffusion (LTHD) module to enhance task-agnostic feature modeling and reduce computational complexity. The authors conducted experiments on PASCAL-Context and NYUD-v2 datasets, demonstrating significant performance gains using the proposed method. The paper is well-organized, clearly presented, and includes comprehensive evaluations that highlight the effectiveness of TIF.

- Summary of Weaknesses and Rebuttal Response

Despite the strengths, several concerns were raised by the reviewers:

1. Novelty and Insight (Reviewer 2zx2): Reviewer 2zx2 noted that the overall network pipeline, consisting of a task-sharing encoder and task-specific decoders, is common in addressing multi-task problems. The main technical contribution, the SAFF module, was seen as lacking deep insight. Despite the authors’ rebuttal explaining the design rationale, the reviewer maintained their position that the technical contribution is incremental. I agree with Reviewer 2zx2 the novelty is not significant, as task-specific module is a common way in multi-task learning. However, I acknowledge the techniqual contribution of the proposed framework, which is simple and effective compared with recent methods (e.g., MLoRE).

2. Negative Transfer and Generalizability (Reviewer nnEZ): Reviewer nnEZ appreciated the novelty in addressing negative transfer but emphasized the need for more clarity on when the impact of negative transfer is severe enough to justify a task-interaction-free approach. The reviewer also requested more details on the generalizability of the proposed method with different backbones and larger kernel sizes. The authors addressed these points in the rebuttal, clarifying the scenarios of negative transfer and providing additional results.

3. Comparison with SOTA Methods (Reviewer k6Rn): Reviewer k6Rn highlighted the need for comparisons with more recent state-of-the-art methods and additional analyses on the method’s effectiveness. The reviewer also pointed out minor issues like missing parameter and FLOP comparisons and a typographical error. The authors’ rebuttal included comparisons with recent methods and detailed analyses, addressing most of these concerns.

- Final Recommendation

After considering the detailed feedback from all reviewers and the authors’ responses, I recommend accepting the paper. The authors have provided sufficient evidence of the method’s effectiveness through extensive experiments and addressed most of the concerns raised during the review process. While there are areas for improvement, particularly in providing deeper insights and broader comparisons, the contributions of this paper are significant and relevant to the multimedia and multimodal processing community.